# Mapping the genealogy of medical device predicates in the United States

**Dhruv B. Pai** *

Science, Mathematics and Computer Science Program, Montgomery Blair High School, Silver Spring, MD, United States of America

* dhruvbpai@gmail.com

## Abstract

### Background

In the United States, medical devices are regulated and subject to review by the Food and Drug Administration (FDA) before they can be marketed. Low-to-medium risk novel medical devices can be reviewed under the De Novo umbrella before they can proceed to market, and this process can be fairly cumbersome, expensive, and time-consuming. An alternate faster and less-expensive pathway to going to market is the 510(k) pathway. In this approach, if the device can be shown to be substantially equivalent in safety and effectiveness to a pre-existing FDA-approved marketed device (or "predicates"), it can be cleared to market. Due to the possibility of daisy-chaining predicate devices, it can very quickly be difficult to unravel the logic and justification of how a particular medical device's equivalence was established. From patients' perspective, this minimizes transparency in the process. From a vendor perspective, it can be difficult to determine the right predicate that applies to their device.

### Methods

We map the connectivity of various predicates in the medical device field by applying text mining and natural language processing (NLP) techniques on data publicly made available by the FDA 78000 device summaries were scraped from the US FDA 510(k) database, and a total of 2,721 devices cleared by the 510(k) regulatory pathway in 2020 were used as a specific case study to map the genealogy of medical devices cleared by the FDA. Cosine similarity was used to gauge the degree of substantial equivalence between two medical devices by evaluating their device descriptions and indications for use. Recalls and complaints for predicate devices were extracted from the FDA's Total Product Life Cycle database using html scraping and web page optical character recognition to determine the similarity between class 1 recalled devices (the most severe form of device recall) and other substantially equivalent devices. A specific product code was used to illustrate the mapping of the genealogy from a De Novo device.

### Results and discussion

The ancestral tree for the medical devices cleared in 2020 is vast and sparse, with a large number of devices having only 1–2 predicates. Evaluation of substantial equivalence data

**Data Availability Statement:** The minimal anonymized dataset necessary to replicate the study findings are available on Kaggle (DOI: 10.34740/kaggle/dsv/1949990).

**Funding:** The author(s) received no specific funding for this work.

**Competing interests:** The authors have declared
that no competing interests exist.

from 2003–2020 shows that the standard for substantial equivalence has not changed significantly. Studying the recalls and complaints, shows that the insulin infusion pump had the highest number of complaints, yet none of the recalled devices bore significant degree of text similarity to currently marketed devices. The mapping from the De Novo device case study was used to develop an ancestry map from the recalled predicate (recalled due to design flaws) to current substantially equivalent products in the market.

## Conclusions

Besides enabling a better understanding of the risks and benefits of the 510(k) process, mapping of connectivity of various predicates could help increase consumer confidence in the medical devices that are currently in the marketplace.

## Introduction

### Medical device industry

As per Fortune Magazine [1], the 2018 global medical devices market size was 425.5Billion USD and is expected to expand to 612.7Bn USD by 2025. The North American market was valued at 169.3Bn USD in 2018. In order to legally market a medical device in the US, the most common forms of premarket submissions to the US Food and Drug Administration (FDA) are the 510(k) premarket notification submission, the PMA premarket approval, and the De Novo pathway. Each of these submission types results in a determination by FDA that clears [510(k)], approves [PMA], or grants [de novo] marketing rights to the successful submitter. A 510(k) is a premarket submission made to FDA to demonstrate that the device to be marketed is at least as safe and effective (substantially equivalent) to a legally marketed device that is not subject to PMA [2] (i.e., a De Novo or another 510(k)).

### Substantial equivalence

Clearance through the 510(k) route is based on demonstrating the "substantial equivalence" of a device to an existing and previously cleared or granted device. According to the US FDA, the legally marketed device(s) to which equivalence is drawn is known as the predicate device(s) [3].

Two critical components for 510(k) clearance are the "indications for use" and "device description". When viewed in public-facing summary statements provided for 510(k)-cleared devices, these two sections are the most apparent indicators of substantial equivalence. The subjective aspects of these two sections can dilute the true intent of the indications for use and/ or device descriptions from the first approved device to successive generations of new devices claiming that first approved device as a predicate. And yet, because the full genealogy of ancestral relationships connecting devices cleared through the 510(k) framework is not reported (only the immediate predicates of a device are listed), these deviations from original content can be hard to track.

One of the key problems previously identified with the 510(k) framework is that despite the legal requirement that the scientific evidence demonstrating substantial equivalence to a predicate device be made publicly available, this information is often either lacking or insufficient [4–8]. On the other hand, even when substantial equivalence to a predicate is demonstrated, this does not ensure the safety of the device, if—as may be the case—the predicates used were

also cleared without sufficient evidence of safety and effectiveness, but rather through the mechanism of substantial equivalence to another predicate [9]. The serious complications suffered by some patients have resulted in a number of class-action lawsuits and criticism that the 510(k) is not providing sufficient protection for patients [5,10–13].

On the other hand, despite the numerous complaints against the substantial equivalence system (SES) and the recurring device problems that arise [14], the 510(k) process is still important for the FDA's objective of protection and promotion of public health [15]. Comparisons between the European and American systems for medical device approval have shown [16] that because of the speed by which medical device approval occurs in the EU as opposed to the US, consumers in the US might have to wait an additional 3 years after their EU counterparts get access to potentially lifesaver medical innovation. However, in the case of the 510(k) substantial equivalence system, this study found there was not a significant difference in the speed of approval between this system and its EU counterpart. In both systems, there is a clear understanding that post-market surveillance and public access to post-market data are essential to ensure the performance, safety, and quality of devices approved by the two regulatory bodies [6,7]. FDA's total product life cycle (TPLC) database [17] is an approach to address some of these concerns, as are frameworks such as IDEAL-D (Idea, Development, Exploration, Assessment, Long term study—for Devices) [18], a model for integrated stepwise evaluation of maturing interventions in surgical procedures.

This manuscript aims to provide a comprehensive view of the evolution of the U.S. FDA 510(k) predicate system, and help track how faithfully the device description and indications for use of a 510(k)-cleared device aligns with its predicate genealogy. There have been attempts to map the regulatory ancestral network for different devices [13,19,20], e.g., tracking the predicate tree for a specific group of surgical meshes approved between 2013 and 2015 [20]. That tracking study analyzed a small section of the 510(k) network but was able to demonstrate that tracing of predicate data as well as analysis of the relationship between recalled and cleared 510(k) devices was possible. Similarly, for a subset of predicates between 2008 and 2012, it was demonstrated that information on substantial equivalence was missing [19]. Notably, a sample of only 50 devices out of the more than 15, 000 approved during this period was used to conduct the study.

## Materials & methods

78000 device summaries (1996-current) were scraped from the US FDA 510(k) predicate database [21] via HTTP requests and text scraping. A total of 2,721 devices cleared by the 510(k) regulatory pathway in 2020 were used as a specific case study to track the genealogy. Through regular expression algorithms and optical character recognition (OCR) in Python, the predicate for each of the devices was found and mapped with the device itself, forming the predicate genealogy.

Within the summaries of cleared medical devices, the sections for device description and indications for use tend to be small (50–100 words each), and hence a method like cosine similarity (CS) [22] suffices for gauging the degree of substantial equivalence of two medical devices. Mathematically, cosine similarity represents the cosine of the angle between two vectors, $\vec{A}$ and $\vec{B}$. It is computed as:

$$cosine\ similarity\ (\vec{A},\ \vec{B}) = \frac{\vec{A} \circ \vec{B}}{\|\vec{A}\| \times \|\vec{B}\|} \tag{1}$$

Any text can be encoded into a vector through bag-of-words embedding. The bag-of-words embedding decomposes a text into the frequency of its individual words [23]. The naïve

embedding of word frequencies for n-words corresponds to an n-dimensional vector, upon which cosine similarity can be performed to compare to another vector. However, this approach can treat two words that are very similar in meaning as distant in feature space.

The Word2Vec model generates a robust representation of text in a low-dimensional space, while ensuring that texts with similar meanings are clustered closely in feature space [24]. The Word2Vec neural network used for the predicate genealogy mapping project was pretrained on all predicate summary texts using the Gensim Python library. The model embedded the original tokens into a 500-dimensional space using a shallow neural network (a neural network with only two layers). The first layer took the bag-of-words frequency embedding as input and then mapped to 500 output neurons which each represented the magnitude of the vector in that dimension. The result was a 500-dimension vector that represented the underlying set of meanings of the text at hand, which could be directly compared to other vectors using cosine similarity. Before the embedding was performed, stop words were removed from the text. Stop words are common words such as "a", "and," or "but" that serve no purpose other than to link concepts together in the text [25]. The method was applied separately to the "indications for use" and the "device description" sections for each predicate summary.

## Testing recalls and complaints for predicate devices

Another important aspect of the project was testing recalls and complaints for predicate devices. To do this, text mining approaches were developed for the FDA database that incorporated both html scraping and web page OCR. For each of the 2721 cleared devices under consideration, the product code was acquired. Then, for each product code, the number of complaints, devices, and list of recalls were mined from the FDA Total Product Life Cycle (TPLC) database [17]. The complaints and device counts were compared to see whether the number of issues correlated with the number of devices and vice versa. Recalls were split into Class I (most severe), II, and III (least severe), and complaints were divided between device issues and consumer complaints. Complaints were projected onto a log scale because a few product codes had very large numbers of complaints. The size of each point corresponds to the number of different devices in that product code. While one can expect a causality between complaints and recalls (i.e. an increase in complaints leads to a recall), it is hard to quantify this relationship as there can be a lag between complaints and when a recall notice is placed, primarily due to the sparsity of details about the timing of the complaints.

For 7 outlier product codes identified by their complaint to device ratio, the list of recalls was analyzed. The class 1 recalls, which are the most severe, and their corresponding 510(k) summaries were compared through text similarity with non-recalled devices in the same class, to see whether any current devices on the market demonstrate substantial equivalence to potentially hazardous recalled devices. A high degree of similarity between a device that prompted a severe recall and a device that is currently being marketed could act as a predictor for whether the marketed device could be faulty or demonstrate potentially fatal issues. When text similarity was compared for these trials, only indications for use and device description were evaluated because these sections are most important in determining substantial equivalence, from a public-facing perspective.

The ProCode, LZG, [26] with the highest number of complaints (8 Class 1, 22 Class 2, and 4 Class 3) was used as a case study. The 510(k) products with class 1 recalls (the most severe category) were compared through text similarity with non-recalled devices in the same Pro-Code category.

### Genealogy traversal from a De Novo device

To determine the genealogy traversal from a *De Novo* device, the ProCode MRN [27], was analyzed as a case study. MRN is the product code for Nitric Oxide administration apparatus, and is classified as a Class II device. Using text mining and the cosine similarity method, the predicate genealogy of all currently marketed medical devices under that product code was traced.

## Results

Fig 1 is a histogram distribution of the branching factor of predicate devices in the ancestral tree of the 2721 devices that had been cleared in 2020. The branching factor for a given device is defined as the number of predicates for that device in the tree, also known as the outdegree in a directed graph. For the devices in the tree for each year, the branching factor for those devices was plotted in the histogram.

Fig 2 traces the genealogy of predicates and substantial equivalence of the 2,721 devices cleared in 2020. Using these devices and their ancestral predicates, which comprised a total of 10,576 devices, the substantial equivalence of a device to its predicates over time was calculated. These results are displayed in Fig 2 as a box-and-whisker plot. Note that the quality of PDF files is relatively poor in the older data (circa 2000–2002); while the OCR approach could extract the device numbers, it was unable to extract the information regarding indications for use and technological characteristics in PDF files from 2000–2002, hence Fig 2 lists substantial equivalence from 2003 onwards.

### Testing recalls and complaints for predicate devices

Fig 3 plots the complaints (since 2006, as the TPLC database retains data for only 15 years) versus recalls for the product codes (ProCodes) associated with the 2,721 devices from 2020. The recalls are broken down into three subcategories, Class III, II, and I with increasing severity respectively. The complaints are also further classified in two ways: events with the device, and patient complaints. The former reflects the number of devices with an adverse manufacturing

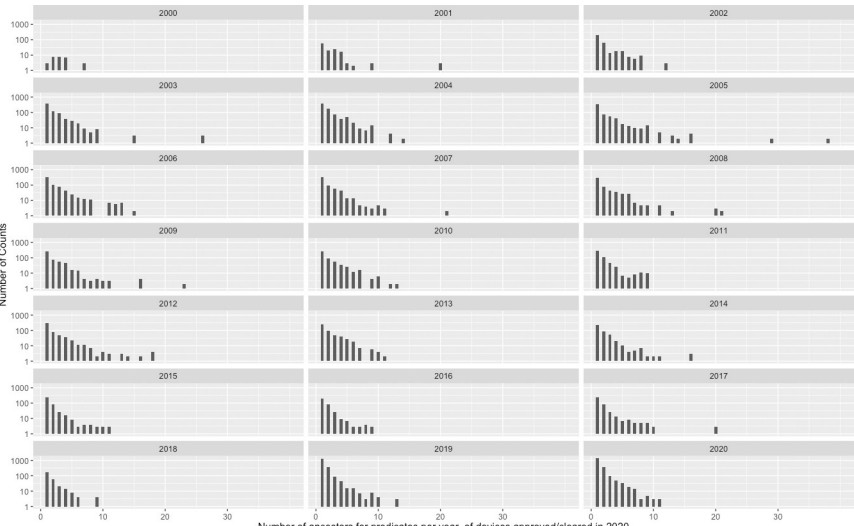

**Fig 1. Histogram showing the distribution of the number of predicate "parent" devices from 2000 to 2020 for 462 medical devices cleared through the 510(k) regulatory pathway in 2020.** The y axis is a log scale. Note that an outlier of 62 predicates (histogram count of 1) in 2015 has been not indicated, in order to enhance readability of the histograms shown.

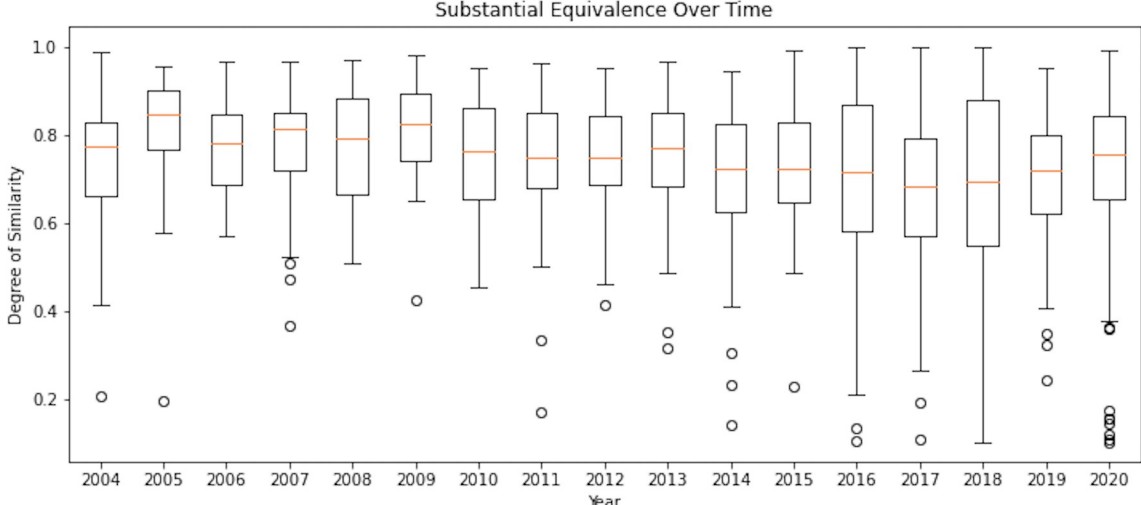

**Fig 2. Box and whisker plots for the average equivalence to predicate device across all approved devices each year, from 2003 onwards.** The degree of similarity index (y-axis) indicates the level of equivalence between a device and its predicate (1: Identical, 0: Totally dissimilar).

defect(s) and the latter reflects the number of adverse events on patients caused by said manufacturing defect(s). Generally, a singular faulty device causes a single negative patient event, though this is not always the case. As a result, there are small, albeit significant, differences between patient and device complaint counts. The size of a circle is proportional to the number of products in that product code.

Amongst the outliers, the ProCode for Insulin Infusion Pumps, LZG, had the highest number of complaints. There have been 611,769 device complaints and 414,024 patient complaints across 60 devices since 2006 and 8 unique Class I recalls (along with 22 Class II and 4 Class III recalls). Fig 4 compares the eight Class 1-recalled devices to 52 currently marketed devices under ProCode LZG using text similarity computed by cosine similarity.

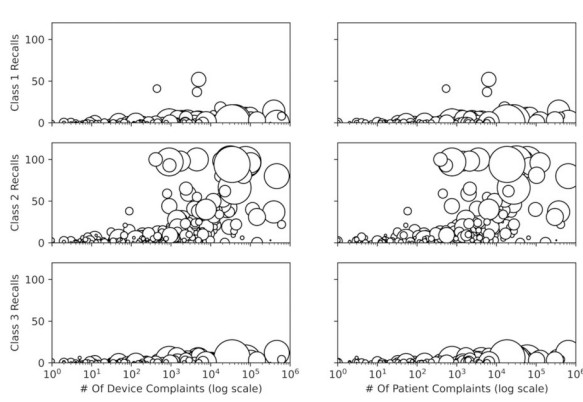

**Fig 3. Scatter plot of device complaints (since 2006, as the TPLC database retains data for only 15 years) versus recalls for the product codes associated with the 2,721 devices from 2020.** Recalls were split into Class I (most severe), II, and III (least severe), and complaints were divided between device issues and consumer complaints. Complaints were projected onto a log scale because a few product codes had very large numbers of complaints. The size of each point corresponds to the number of different devices in that product code.

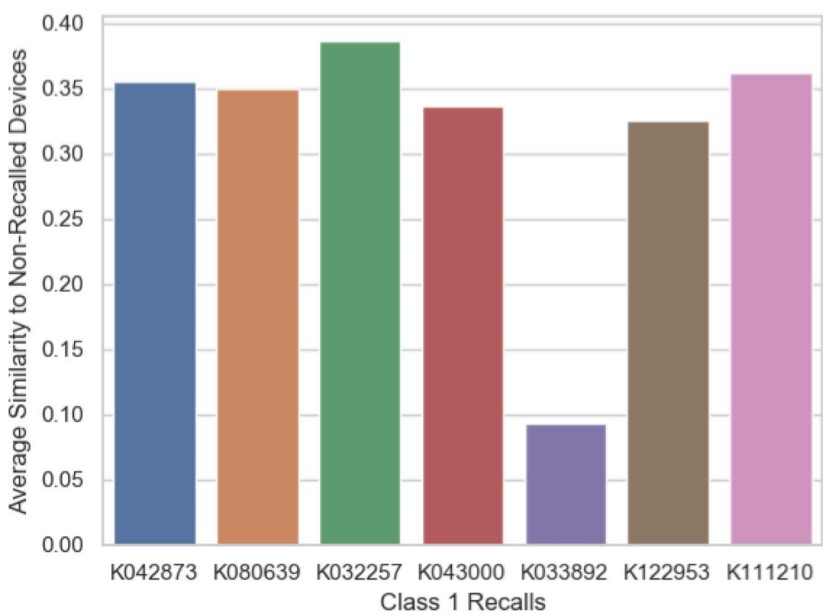

**Fig 4. Bar plot comparing seven Class 1 recalls to 87 predicate devices, all under the LZG ProCode.** The results are based on text similarity computed by the cosine similarity algorithm.

## Genealogy traversal from a De Novo device

The Sankey diagram in Fig 5 maps the predicate tree of the MRN ProCode, (the product code for Nitric Oxide administration apparatus) with the thickness of the edge connecting the device nodes representing the similarity value (thus, a thinner edge implies poor similarity). For the ProCode MRN, all currently marketed devices are found to originate from the medical

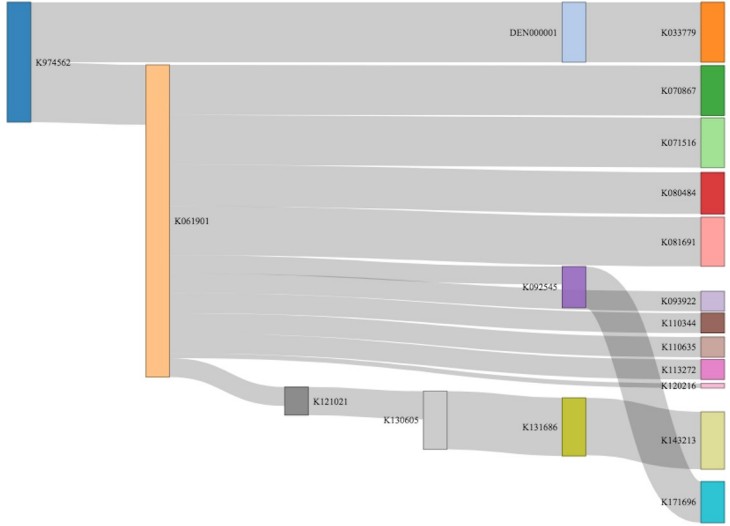

**Fig 5. Sankey plot showing the genealogy of medical devices cleared through the 510(k) approach for a De Novo product DEN000001 (which was initially cleared as K974562 and reclassified to a De Novo product with special controls).** The thickness of the connecting edge corresponds to the similarity match between a predicate and a successive device (the exceptions being the incoming/outgoing connections for the product K033779, for which summary statements are not available).

device K974562, which was reclassified as a De Novo device, DEN000001, with special controls. Table 1 shows the similarity values for K061901 and devices descendant from the predicate ancestral tree of K061901; K061901 inherits from the predicate K974562/DEN000001.

## Discussion

The ancestral tree for the medical devices cleared in 2020 is vast and yet sparse. While the network can reach out to the earliest dates of the database, a large number of devices have 1–2 predicates only. If one assumes text similarity to be a good indicator of substantial equivalence approval, Fig 2 may suggest that the standard for substantial equivalence has not changed over time when the data from 2003-onwards is considered.

We assume here that an ideal product code would have many devices, few complaints, and few recalls as this combination implies a reliable device/predicate family with a high degree of customer satisfaction (and potentially a high level of safety and effectiveness). Fig 3 shows that a majority of the product codes had very few devices, and few complaints of both varieties (device and patient), and minimal Class 1 and 3 recalls, with Class 1 being the most severe recalls. There were a greater number of product codes with many class 2 recalls, suggesting that this type of recall is more prevalent. Class 1 and class 3 recalls were not found to have a strong association with the number of complaints in either regard. As shown by the size of the points in Fig 3, product codes which had a high number of class 2 recalls generally had more devices (point size corresponds to the number of different devices in that product code) and also more complaints compared to product codes with fewer class 2 recalls.

The results plotted in Fig 4 demonstrate that none of the recalled devices under the product code LZG, bear any significant degree of text similarity (average text similarity $< 0.5$) to currently marketed devices. This may suggest that the current products under this ProCode are potentially safer than the recalled devices and likely not susceptible to the same issues that prompted the class 1 recalls. Obviously, this does not exclude the possibility that the newer device may get recalled for a totally different issue.

**Table 1. Similarity score for devices descendant from the predicate ancestral tree for the ProCode: MRN.**

| Predicate | Descendant Device | Similarity |
|---|---|---|
| K974562 | DEN000001 | N/A |
| DEN000001 | K033779 | N/A |
| K974562 | K061901 | N/A |
| K061901 | K070867 | 0.83 |
| K061901 | K071516 | 0.83 |
| K061901 | K080484 | 0.70 |
| K061901 | K081691 | 0.82 |
| K061901 | K092545 | 0.30 |
| K061901 | K093922 | 0.33 |
| K061901 | K110344 | 0.33 |
| K061901 | K110635 | 0.34 |
| K061901 | K113272 | 0.33 |
| K061901 | K120216 | 0.08 |
| K061901 | K121021 | 0.32 |
| K121021 | K130605 | 0.47 |
| K130605 | K131686 | 0.97 |
| K131686 | K143213 | 0.96 |
| K092545 | K171696 | 0.69 |

For the ProCode MRN, both the originating medical device K974562 and its sister product, K033779, had class II recalls due to design flaws. Despite its class II recall, K974562 is the predicate ancestor for every currently marketed device under the MRN ProCode through its descendant device, K061901. As the publicly available summary statements are only available for the predicate K061901 and descendant devices, the cosine similarity approach was used to determine the similarity value for K061901 and descendant devices, as shown in Table 1. K061901 is the direct ancestor of 11 other medical devices.

There are a number of limitations to this study that we hope to address in future iterations. We were only able to analyze predicates approved after 2002 because of limitations in OCR technology. Predicate summaries of devices approved before 2002 yielded significant errors in text scraping, resulting in uninterpretable erroneous unicode characters. There were a number of potential causes for these errors, including rotated and offset pages. Improvements in OCR technology combined with pdf preprocessing could improve scraping for early predicates. The Word2Vec model we trained on predicate summaries can only generate vectors for words it has seen before. As a result, words that were not used in training of the model cannot be interpreted. This is problematic because of the diversity of jargon even across small families of predicates, let alone the entire predicate genealogy. Using our Word2Vec model to generate a bag of words for different predicates that were not included in this study could yield high errors, because much of the device-specific technical vocabulary may not be recognized. Finally, and most importantly, the results of our approach require a public interface for visualization and analysis. In order to address a lack of transparency in the medical device approval system, stakeholders should be able to inspect medical device approvals at both a fine resolution (one predicate family) and a large scale (entire predicate network for a year). In the future, we hope to develop a public-facing, open-source website where consumers, manufacturers, and regulatory agents alike can interact with the predicate network in real-time.

## Conclusions

Based on the analysis of the indications for use and the technological features of the various predicate-descendant medical device combinations, the predicate ancestral tree can be shown to develop as a branching pattern. Though the predicate system has evolved extensively since its deployment, the standard for substantial equivalence does not appear to have changed. New product approvals under product codes which have been associated with many severe recalls do not appear to be substantially equivalent to the recalled products.

## Author Contributions

**Conceptualization:** Dhruv B. Pai.

**Data curation:** Dhruv B. Pai.

**Formal analysis:** Dhruv B. Pai.

**Investigation:** Dhruv B. Pai.

**Methodology:** Dhruv B. Pai.

**Project administration:** Dhruv B. Pai.

**Resources:** Dhruv B. Pai.

**Software:** Dhruv B. Pai.

**Supervision:** Dhruv B. Pai.

**Validation:** Dhruv B. Pai.

**Visualization:** Dhruv B. Pai.

**Writing – original draft:** Dhruv B. Pai.

**Writing – review & editing:** Dhruv B. Pai.

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
