## [Decision Letter · Decision Letter 0]

6 Jul 2021

PONE-D-21-19490

Mapping the genealogy of medical device predicates in the United States

PLOS ONE

Dear Dr. Pai,

Thank you for submitting your manuscript to PLOS ONE. After careful consideration, we feel that it has merit but does not fully meet PLOS ONE’s publication criteria as it currently stands. Therefore, we invite you to submit a revised version of the manuscript that addresses the points raised during the review process.

We look forward to receiving your revised manuscript.

Kind regards,

Dylan A Mordaunt

Academic Editor

PLOS ONE

Additional Editor Comments:

I think there is value in this paper. It needs some refinement both to increase the appeal and relevance to a broader audience, but also to increase the potential impact. Our reviewers have provided some good suggestions that the authors should consider and address. In taking on a complex cross-disciplinary area, they have done well but created a considerable challenge in terms of creating a meaningful output so far for anyone other than a highly technical (and therefore highly limited) range of readers. Please reconsider your paper with regards to the range of authors to whom this is both of potential interest and potential application. Thanks again, Dylan.

Journal Requirements:

Reviewers' comments:

Reviewer's Responses to Questions

**Comments to the Author**

1. Is the manuscript technically sound, and do the data support the conclusions?

Reviewer #1: Partly

Reviewer #2: Yes

2. Has the statistical analysis been performed appropriately and rigorously? 

Reviewer #1: Yes

Reviewer #2: N/A

3. Have the authors made all data underlying the findings in their manuscript fully available?

Reviewer #1: Yes

Reviewer #2: Yes

4. Is the manuscript presented in an intelligible fashion and written in standard English?

Reviewer #1: No

Reviewer #2: Yes

5. Review Comments to the Author

Reviewer #1: the use of predicate / child and predicate / children terms is confusing. suggest chage to antecedent / sucessor or similar. The discussion section includes statements that are not supported with data that is obvious eg "prodiuct codes with more class 2 recalls tended to have more devices and more complaints" is not supported by data and analysiss within the body of the paper. similarly the conlcusions are unsupported by data or anlsyis from the body of the paper - for example the statement that ".. the technological characteristics of the predicate and its child can fluctuate wiedly, suggetsing a wide vaiation in the accpeted characteristics of the substantial equivalence test."

both the absence of supprting data and the volubl language will need to be adressed.

Reviewer #2: Dhruv High Pai summarizes in this article that if the medical devices can be shown to be substantially equivalent in safety and effectiveness to a pre-existing FDA via the 510(k) pathway, they can be cleared to market. Author claims that mapping of connectivity of various predicates could help increase consumer confidence in the medical devices that are currently in the marketplace. Devices approved first in the EU are associated with an increased risk of post-marketing safety alerts and recalls, therefore the paper addresses an essential public health concern and this is a strength to acknowledge. The study appears to be sound, however some details should be expanded and clarified to ensure that readers understand what the researcher studied.

Some points to consider in subsequent versions:

Major – The study lacks general statement about its limitations. However, the use of bag-of-words model as one of the most popular representation methods for object categorization is associated with a number of uncertainties. The key idea is to quantize each extracted key point into one of visual words, and then represent each image by a histogram of the visual words. For this purpose, a clustering algorithm (e.g., K-means), is generally used for generating the visual words. As K-means tends to ‘starve’ medium density regions in feature space, measures to minimize the loss of information should be implemented and justified. It has been previously demonstrated that the pairwise similarity cannot be computed directly and to show the statistical consistency, Kernel Density Function Estimation / also referred as “Quantization via Kernel Estimation”, or QKE for short/ is encouraged in previous similar researches as an alternative to empirical distribution as “Quantization via Empirical Estimation”, or QEE. Therefore, additional justification is needed how author avoids in the model the problem of ‘code word uncertainty’ and ‘code word plausibility’.

Major – the references are very limited and only 8 /15 in total/ are research articles. The 510(k) ancestry as well as the related safety and effectiveness concerns have been elaborated in a number of rigorously peer- review publications, to note a few:

Ardaugh BM, Graves SE, Redberg RF (2013) The 510(k) ancestry of a metal-on-metal hip implant. N Engl J Med 368: 97–100.

Kramer DB, Xu S, Kesselheim AS (2012) Regulation of medical devices in the United States and European Union. N Engl J Med 366: 848–855

Zuckerman D, Brown P, Das A (2014) Lack of publicly available scientific evidence on the safety and effectiveness of implanted medical devices. JAMA Intern Med 174: 1781–1787.

Zuckerman DM, Brown P, Nissen SE (2011) Medical device recalls and the FDA approval process. Arch Intern Med 171: 1006–1011

Kessler DA, Pape SM, Sundwall DN (1987) The Federal regulation of medical devices. N Engl J Med 317: 357–366.

Day CS, Park DJ, Rozenshteyn FS, Owusu-Sarpong N, Gonzalez A (2016) Analysis of FDA-Approved Orthopaedic Devices and Their Recalls. J Bone Joint Surg Am 98: 517–524.

Fargen KM, Frei D, Fiorella D, McDougall CG, Myers PM, Hirsch JA, et al. (2013) The FDA approval process for medical devices: an inherently flawed system or a valuable pathway for innovation? J Neurointerv Sur 5: 269–275.

Haylen BT, Sand PK, Swift SE, Maher C, Moran PA, Freeman RM. (2012) Transvaginal placement of surgical mesh for pelvic organ prolapse: more FDA concerns—positive reactions are possible. Int Urogynecol J 23: 11–13.

Sedrakyan A, Campbell B, Merino JG, Kuntz R, Hirst A, McCulloch P. (2016) IDEAL-D: a rational framework for evaluating and regulating the use of medical devices. BMJ 353: i2372.

Sorenson C, Drummond M (2014) Improving medical device regulation: The United States and Europe in perspective. Milbank Q 92: 114–150.

I hereby encourage the author to enrich his references as it is hard to explain why such landmark sources were omitted.

Minor – more robust explanations are needed to explain the impact of poor quality of PDF files /line 167 onwards/ and the limitations of the OCR /lines 168-171/ to verify the efficacy of the proposed framework by applying it to object recognition.

Minor- In testing recalls and complaints for predicate device /lines 176- 199/ - may author elaborate on how many complaints have been found causal.

In summary, the paper raises interesting points that seem worth discussing, but requires

revisions.

6. PLOS authors have the option to publish the peer review history of their article (what does this mean?). If published, this will include your full peer review and any attached files.

Reviewer #1: No

Reviewer #2: **Yes: **Borislav Borissov

---

## [Author Response · Author response to Decision Letter 0]

5 Sep 2021

Reviewer #1: the use of predicate / child and predicate / children terms is confusing. suggest chage to antecedent / sucessor or similar. 

Author Response: I agree that the terminology can be confusing. Since the predicate term is a defined term in regulatory use, and is used by publications that have been tracking medical device safety and effectiveness issues (such as the references cited by the reviewers), I provided a definition of the “predicate” word to stay true to that usage. Additionally, instead of using the “children” term, I steered clear of that language; instead using ancestral tree of the predicate (this is how other publications referred to by the reviewers define them), and brought in the word “descendant” to describe the devices that come from the predicate’s ancestral tree. I minimized the use of the predicate-child relation language as much as possible; I do hope this makes the terminology clearer.

Reviewer #1: The discussion section includes statements that are not supported with data that is obvious eg "prodiuct codes with more class 2 recalls tended to have more devices and more complaints" is not supported by data and analysiss within the body of the paper. 

Author Response: I thank the reviewer for pointing this out. I have added text in the discussion to provide more clarity about Figure 3, and how the size of the data points visualized is proportional to the number of different devices in a particular product code. I hope that the added/amended text in the discussion section clarifies this point.

Reviewer #1: similarly the conlcusions are unsupported by data or anlsyis from the body of the paper - for example the statement that ".. the technological characteristics of the predicate and its child can fluctuate wiedly, suggetsing a wide vaiation in the accepted characteristics of the substantial equivalence test." both the absence of supprting data and the volubl language will need to be adressed.

Author Response: I agree with the reviewer that the statement in the conclusion that the reviewer points out is not supported by data in the manuscript. The corresponding text in the conclusion has been deleted.

Reviewer #2: Major – The study lacks general statement about its limitations. However, the use of bag-of-words model as one of the most popular representation methods for object categorization is associated with a number of uncertainties. The key idea is to quantize each extracted key point into one of visual words, and then represent each image by a histogram of the visual words. For this purpose, a clustering algorithm (e.g., K-means), is generally used for generating the visual words. As K-means tends to ‘starve’ medium density regions in feature space, measures to minimize the loss of information should be implemented and justified. It has been previously demonstrated that the pairwise similarity cannot be computed directly and to show the statistical consistency, Kernel Density Function Estimation / also referred as “Quantization via Kernel Estimation”, or QKE for short/ is encouraged in previous similar researches as an alternative to empirical distribution as “Quantization via Empirical Estimation”, or QEE. Therefore, additional justification is needed how author avoids in the model the problem of ‘code word uncertainty’ and ‘code word plausibility’.

Author Response: I thank the reviewer for indicating that the study lacks general statements about its limitations. I have added a section in the discussion portion indicating the limitations in the current study, as well as the Word2Vec model that I used for this work. I am unsure about the latter point raised by the reviewer, regarding visual words. I believe visual words are used for extracting information content from images (e.g. “cat”, “dog”, from an image containing a cat and/or a dog), and the QKE and QEE approaches apply in that context. However, the current project does not use visual words and does not extract image information in that manner, and hence the visual word issues like ‘code word uncertainty’ and ‘code word plausibility’ would not apply to the current work. I do apologize if this did not come across clearly in my manuscript, and I have added more content in the Materials and Methods section providing details about the Word2Vec model that I used for this project. I hope that this clarifies the method used to develop the mapping. 

Reviewer #2: the references are very limited and only 8 /15 in total/ are research articles. The 510(k) ancestry as well as the related safety and effectiveness concerns have been elaborated in a number of rigorously peer- review publications,< a number of publications are provided, but not shown here for the sake of brevity>

Author’s Response: I thank the reviewer for pointing this significant weakness in the manuscript, and appreciate the references that are provided. These have now been added in the appropriate sections of the background content of the manuscript, and I believe have strengthened the quality of the manuscript considerably. 

Reviewer #2: more robust explanations are needed to explain the impact of poor quality of PDF files /line 167 onwards/ and the limitations of the OCR /lines 168-171/ to verify the efficacy of the proposed framework by applying it to object recognition.

Author’s Response: Explanation is provided in the discussion section for why OCR use for text extraction yielded significant errors for PDF documents from before 2002. 

Reviewer #2: In testing recalls and complaints for predicate device /lines 176- 199/ - may author elaborate on how many complaints have been found causal.

Author’s Response: The suggestion from the reviewer is an excellent one; however, the TPLC database provided by the FDA does not have temporal resolution within the complaints to allow the establishment of causal relation between the product recalls and the complaints. I have made a mention of this point in the discussion section of the manuscript.

---

## [Decision Letter · Decision Letter 1]

20 Sep 2021

Mapping the genealogy of medical device predicates in the United States

PONE-D-21-19490R1

Dear Dr. Pai,

We’re pleased to inform you that your manuscript has been judged scientifically suitable for publication and will be formally accepted for publication once it meets all outstanding technical requirements.

Kind regards,

Dylan A Mordaunt

Academic Editor

PLOS ONE

Additional Editor Comments (optional):

Thank you for your resubmission. The revised manuscript has been assessed by the reviewers and there is agreement to accept without any further revisions.

Reviewers' comments:

Reviewer's Responses to Questions

**Comments to the Author**

1. If the authors have adequately addressed your comments raised in a previous round of review and you feel that this manuscript is now acceptable for publication, you may indicate that here to bypass the “Comments to the Author” section, enter your conflict of interest statement in the “Confidential to Editor” section, and submit your "Accept" recommendation.

Reviewer #1: All comments have been addressed

Reviewer #2: All comments have been addressed

2. Is the manuscript technically sound, and do the data support the conclusions?

Reviewer #1: Yes

Reviewer #2: Yes

3. Has the statistical analysis been performed appropriately and rigorously? 

Reviewer #1: Yes

Reviewer #2: Yes

4. Have the authors made all data underlying the findings in their manuscript fully available?

Reviewer #1: Yes

Reviewer #2: Yes

5. Is the manuscript presented in an intelligible fashion and written in standard English?

Reviewer #1: Yes

Reviewer #2: Yes

6. Review Comments to the Author

Reviewer #1: thank you for addressing my comments. the changes meet the issues raised in my first comments., including conclauions based on data and information provided in the body.

Reviewer #2: The author has embedded in the appropriate sections of the background content of the manuscript the reviewers comments and have strengthened the quality of the manuscript appropriately.

7. PLOS authors have the option to publish the peer review history of their article (what does this mean?). If published, this will include your full peer review and any attached files.

Reviewer #1: No

Reviewer #2: **Yes: **Borislav Borissov

---

## [Editor Report · Acceptance letter]

28 Sep 2021

PONE-D-21-19490R1 

Mapping the genealogy of medical device predicates in the United States 

Dear Dr. Pai:

I'm pleased to inform you that your manuscript has been deemed suitable for publication in PLOS ONE. Congratulations! Your manuscript is now with our production department. 

Kind regards, 

on behalf of

Dr. Dylan A Mordaunt 

Academic Editor

PLOS ONE